# An Efficient and Accurate Multi-Sensor IF Estimator Based on DOA Information and Order of Fractional Fourier Transform

**DOI:** 10.3390/e24040452

**Published:** 2022-03-25

**Authors:** Nabeel Ali Khan, Sadiq Ali, Kwonhue Choi

**Affiliations:** 1Faculty of Engineering and IT, Foundation University Islamabad, Rawalpindi 46000, Pakistan; nabeel.ali@fui.edu.pk; 2Department of Electrical Engineering, University of Engineering and Technology, Peshawar 25000, Pakistan; sadiqali@uetpeshawar.edu.pk; 3Department of Information and Communicaiton, Yeungnam University, Gyeongsan 38541, Korea

**Keywords:** time-frequency, instantaneous frequency, spatial information, angle of arrival

## Abstract

Instantaneous frequency in multi-sensor recordings is an important parameter for estimation of direction of arrival estimation, source separation, and sparse reconstruction. The instantaneous frequency estimation problem becomes challenging when signal components have close or overlapping signatures and the number of sensors is less than the number of sources. In this study, we develop a computationally efficient method that exploits the direction of the IF curve in addition to the angle of arrival as additional features for the accurate tracking of IF curves. Experimental results show that the proposed scheme achieves better accuracy compared to the-state-of-art method in terms of mean square error (MSE) with a slight increase in the computational cost, i.e., the proposed method achieves MSE of −50 dB at the signal to noise ratio of 0 dB whereas the existing method achieves the MSE of −38 dB.

## 1. Introduction

In many real-life scenarios, a signal is acquired through multiple sensors, e.g., electrocardiogram (ECG) signals, electroencephalogram (EEG) signals, radars, and sonars. Most of such signals are non-stationary, i.e., their spectrum changes with time. An amplitude modulation–frequency modulation (AM-FM) is an effective approach to represent such non-stationary signals. An instantaneous frequency is a key parameter for modeling non-stationary signals as signal energy is concentrated along the instantaneous frequency curves in the joint time-frequency (TF) domain. In multi-sensor recordings accurate estimation of the instantaneous frequency is important for a large range of applications including direction of arrival estimation (DOA) [1,2,3], de-noising, blind source separation [4,5,6], and sparse reconstruction [7].

A number of instantaneous frequency estimation has been developed for mono-sensor recordings that include RANSAC-based methods [8,9,10,11], Hough transform-based methods [12,13], Viterbi-based methods [14,15,16,17], image-processing techniques [3,18], ridge detection, and tracking approaches [19,20,21,22]. Most of the above-mentioned methods first transform a given signal to a joint TF domain using time-frequency distributions (TFDs) and then estimate the IF curve by detecting and linking peaks. So, the resolution and robustness to noise of underlying TFD are important for the accurate estimation of IF curves. The resolution of TFDs can be improved using post-processing methods such as adaptive directional kernels [23] and reassignment methods [24,25].

Discrete polynomial transform and fractional Fourier transform are alternative approaches to estimate the parameters of frequency-modulated signals as discussed in [26,27,28]. However, these methods are restricted to linear frequency-modulated chirps only.

In the case of multi-sensor recordings, the IF can be estimated by first separating signal components using blind source separation methods and then estimating the IF of each component separately using mono-sensor methods. The signal components can be separated using multi-variate empirical mode decomposition approaches or synchrosqueezing-based methods [29,30]. However, these methods are only applicable to signals with non-overlapping TF signatures [29,30]. Recently, multi-channel decomposition methods based on Eigen decomposition of auto-correlation matrix and energy concentration measures have been developed to separate signals with highly overlapping TF signatures [31,32]. However, these methods require the number of sensors to be greater or equal to the number of sources [31]. Spatial TFDs present an alternative approach for estimating the IF of multi-component signals in a multi-sensor environment [3,17]. However, all the aforementioned methods are computationally expensive.

In our earlier work, a computationally efficient and robust instantaneous frequency estimation algorithm for multi-sensor recordings was developed for the DOA estimation that outperformed spatial TFD-based methods [33]. The algorithm exploits the rotation order of the fractional Fourier windows as an additional feature for accurate tracking of the IF curve. The computational cost of the algorithm was further reduced by exploiting the slow variation in the IF curve in a separate study, i.e., [2]. It was demonstrated that for IF curves with slow variation, the computational cost can be significantly reduced, without much degradation in accuracy, by only computing IF on a few selected TF points and filling the remaining points through interpolation operation [2]. However, both these methods are based on the assumption that the signal components have significantly different ridge orientation in the region of intersection [2,33].

In this study, we aim to improve the accuracy of our earlier algorithms by exploiting both the DOA information in addition to the direction of ridges for tracking the IF curves. It is demonstrated that the proposed method achieves better results when the angle of intersection between the IF curves of multi-component signals is not very large.

The highlights of this study are:An efficient non-parametric IF estimator is developed for multi-sensor recordings that does not assume that IF of the signal follows any mathematical expression.An efficient non-parametric IF estimator is developed for multi-sensor recordings that do not assume that IF of the signal follows any mathematical expression.For accurate tracking of the IF curve, in addition to the direction of ridges, the proposed estimator exploits additional information of direction-of-arrival provided by multiple sensors. The proposed estimator is developed based on the observation that signal components emitted by different sources with different angles of arrivals having overlapping TF signatures can become non-overlapping in the time-frequency–spatial-frequency domain.The method is applicable both in under-determined and over-determined scenarios.The proposed method is computationally efficient compared to TF-based methods, as the proposed method does not require computation of TFDs of the multi-sensor recordings.

Table 1 illustrates the utility of the proposed method in comparison with the state of art.

The organization of the remaining paper is as follows. In Section 2, the signal model for the proposed work is presented. Section 3 presents details of the methodology of the proposed IF estimation scheme. To assess the performance of the proposed IF estimation scheme, numerical results are presented in Section 4, and work is concluded in Section 5.

## 2. Signal Model

Let us consider a scenario where signals emitted by multiple sources are received by multiple sensors in a uniform linear array as: (1)$$\begin{array}{cc} x_{m}\left(t\right) & =\sum_{k=1}^{K}s_{k}\left(t\right)e^{j\omega_{k}m}\\ \; & =\sum_{k=1}^{K}s_{k}\left(t\right)e^{j2\pi m\left(\frac{d}{\lambda }\right)cos\left(\theta_{k}\right)} \end{array}$$
where $\omega_{k}=2\pi \left(\frac{d}{\lambda }\right)cos\left(\theta_{k}\right)$ is the spatial frequency along sensor axis, i.e., *m*, *d* is inter-sensor spacing, is half of wavelength, $s_{k}\left(t\right)$ is the signal emitted by *k*-th source, *M* is a number of sensors, and *K* is number of sources. We assume that $s_{k}\left(t\right)$ is an AM-FM signal given as [34]:(2)$$s_{k}\left(t\right)=a_{k}\left(t\right)e^{j\Phi_{k}\left(t\right)},$$
where $\Phi_{k}\left(t\right)$ is the instantaneous phase and parameter $a_{k}\left(t\right)$ denotes instantaneous amplitude of the signal. The instantaneous frequency is given as:(3)$$f_{k}\left(t\right)=\frac{d}{dt}\Phi_{k}\left(t\right)$$

## 3. The Proposed Algorithm

In this section, we develop a novel method of tracking instantaneous frequency curves in the joint TF domain by exploiting the additional spatial information provided by multiple sensors in terms of spatial frequency. Joint TF representation of a signal can be obtained using short-time Fourier transform as:(4)$$\rho_{m}\left(t,f\right)=\int x_{m}\left(\tau \right)w\left(t-\tau \right)e^{-j2\pi f\tau }d\tau$$
where w(t) is the analysis window.

### 3.1. Concept of the Proposed Algorithm

As mentioned above, the proposed method is based on the observation that signal components emitted by different sources with different angles of arrival having overlapping TF signatures can become non-overlapping in the time-frequency–spatial-frequency domain. The time-frequency-spatial-frequency domain is obtained by the 2D Fourier transform operation.
(5)$$\rho \left(t,f,\omega \right)=\sum_{m=0}^{M-1}\rho_{m}\left(t,f\right)e^{-j\omega m}$$

Figure 1 illustrates TF representation of a two component signal with overlapping TF representation. This signal becomes non-overlapping when analyzed in time-frequency–spatial-frequency domain, i.e., ρ(t,f,ω), as shown in Figure 2.

### 3.2. Implementation of the Proposed Algorithm

The proposed algorithm first estimates the location of the strongest TF point, then both the direction of the IF curve as well as spatial frequency are found at the strongest TF point. The information provided by fractional Fourier windows as well as spatial frequency is then exploited for tracking the strongest IF curve. Once the IF has been estimated, the corresponding source is removed from $x_{m}\left(t\right)$. The process is iterated till the IFs of all the components have been estimated. The implementation process of the proposed algorithm is illustrated in Figure 3 and details of the main steps are given as follows.

To estimate the strongest TF point, we first estimate the strongest time instant by maximizing the local signal energy as [33]:(6)$$t_{0}=\underset{t}{\arg\mathrm{ma}x}\left(\sum_{m=0}^{M-1}\int_{t-\nabla }^{t+\nabla }{\left|x_{m}\left(t\right)\right|}^{2}dt\right)$$
where −∇to∇ represents time-duration, where energy is computed.

After finding the time instant of the strongest energy point, $t_{0}$, a set of fractional Fourier Gaussian windows is employed to localize a signal around $t_{0}$. The strongest frequency $f_{0}$, optimal rotation order of the Fractional Fourier window $\alpha_{0}$, and the spatial frequency $\omega_{0}$ corresponding to the strongest source are estimated through a 2-dimensional Fourier transform operation as:(7)$$\left(\omega_{0},\alpha_{0},f_{0}\right)=\underset{\left(\omega ,\alpha ,f\right)}{\arg\max}\sum_{m=0}^{M-1}\int \left(w_{\alpha }\left(\tau -t_{0}\right)x_{m}\left(t\right)e^{-j2\pi ft}\right)e^{-j\omega m}d\tau ,$$
where $w_{\alpha }\left(t\right)$ is the fractional Fourier Gaussian window and can be expressed as [35]: (8)$$\begin{array}{cc} w_{\alpha }\left(t\right) & =\frac{e^{j\alpha /2}}{\sqrt{j\sin(\alpha )}}\int_{-\infty }^{\infty }e^{-\frac{\mu^{2}}{2\sigma^{2}}}e^{j\pi (\left(\mu^{2}+t^{2}\right)\cos\left(\alpha -2t\mu \right)/\sin\alpha }d\mu \end{array}$$

In Equation (Equation 8), α=−1,…,−2/L,−1/L,0,1/L,2/L,…,1 and *L* is the number of quantization levels. The $f_{0}$ is the IF of the *k*-th source at $t_{0}$. First we estimate IF for the case where $t>t_{0}$. For this case *t* is incremented as $t=t_{0}+\frac{1}{f_{s}}$ and IF is estimated as:(9)$$\left(\alpha_{0},f_{k}\left(t\right)\right)=\underset{\left(\alpha ,f\right)}{\arg\max}\sum_{m=0}^{M-1}\int \left(w_{\alpha }\left(\tau -t\right)x_{m}\left(t\right)e^{-j2\pi ft}d\tau \right)e^{-j\omega_{0}m},$$
where $\alpha_{0}-1/L\leq \alpha \leq \alpha_{0}+1/L$ and $f_{k}\left(t-1/fs\right)+\nabla \leq f\leq f_{k}\left(t-1/fs\right)+\nabla$. To ensure that the algorithm does not switch to the wrong path, we limit the search space of α and *f* in a limited narrow region around a previously estimated rotation order and peak frequency, i.e., $\alpha_{0}$ and $f_{k}\left(t-1/fs\right)$, because at the intersecting interval the direction of the IF curves will have different directions [2]. The local adaptation of the order of fractional Fourier window along the direction of IF curves ensure that the chirp rate of the analysis window matches with the chirp rate of the component being tracked, thus avoiding destructive interference of components in the intersecting region.

In addition, we also exploit that all the TF points belonging to the same source should have the same direction of arrival, i.e., $\theta_{k},$ that results in the same spatial frequency, i.e., $\omega_{k}$. So, we correlate $e^{-j\omega_{0}m}$ with $x_{m}\left(t\right)$ along the sensor axis to maximize Equation (Equation 9) only for those TF points that correspond to the source that is currently being tracked. Note that in our earlier study, the estimation of the strongest TF point and tracking of the IF curve was done using the following equation [2,33]:(10)$$\left(\alpha_{0},f_{k}\left(t\right)\right)=\arg\max\sum_{m=0}^{M-1}\left|\int w_{\alpha }\left(\tau -t\right)x_{m}\left(t\right)e^{-j2\pi ft}d\tau \right|$$

By comparing Equation (Equation 10) with Equations (Equation 9) and (Equation 7), it is observed that in the earlier work the correlation of $e^{-j\omega_{0}m}$ with x(t,m) was not performed rather simple spatial averaging was performed.

## 4. Results and Discussion

### 4.1. Performance Comparison

The performance of the proposed IF estimation algorithm is compared with the FAST-IF estimation algorithm in [2,35] for both linear and non-linear frequency-modulated signals. For all the examples we employ the mean square error (MSE) as a performance metric. We estimate the MSE curves for signal-to-noise ratio (SNR) ranging from −10 dB to 10 dB by performing 500 simulations.

#### 4.1.1. Sources Emitting Linear Frequency-Modulated Signals

Let us consider signals, in a scenario where linear frequency-modulated signals are intersected by pure tones in the time-frequency domain. The signals emitted by 4 sources are given as:$$\begin{array}{cc} s_{1}\left(t\right) & =e^{j0.1\pi t+j0.001\pi t^{2}}\\ s_{2}\left(t\right) & =e^{j0.22\pi t}\\ s_{3}\left(t\right) & =e^{j0.6\pi t+j0.001\pi t^{2}}\\ s_{4}\left(t\right) & =e^{j0.72\pi t} \end{array}$$
where 0≤t<128 and sampling frequency is 1 Hz. The sources are placed at angles 0°, 10°, 20° and 30°. The TF representations of the received signals are shown in Figure 4 and corresponding IF curves are shown in Figure 5.

These signals are received by 8 sensors. The MSE curves shown in Figure 6 illustrates that the proposed method achieves better performance than the existing method [2].

#### 4.1.2. Sources Emitting Both Linear and Non-Linear Frequency-Modulated Components

Let us consider now consider a scenario where sources emit non-linear frequency modulated signals. We assume signals emitted by 5 sources are given as:$$\begin{array}{cc} s_{1}\left(t\right) & =e^{j0.1\pi t+at^{3}}\\ s_{2}\left(t\right) & =e^{j0.8\pi t}\\ s_{3}\left(t\right) & =e^{j0.1\pi t}\\ s_{4}\left(t\right) & =e^{j0.1\pi t-at^{3}}\\ s_{5}\left(t\right) & =e^{j0.5\pi t} \end{array}$$
where $a=4.0690\times 10^{-6}$. We assume that signal duration is from 0 to 128 s and signal is sampled at 1 Hz. The sources are placed at angles 0°, 10°, 20°, 30° and 40°. The IF curves and TF representations of these signals are shown in Figure 7 and Figure 8 respectively.

For the under-determined scenario, we assume that signals are received by 8 sensors and for the over-determined scenario we assume that we have 4 sensors. The mean square error (MSE) between the estimated IF and the original IF is used as a performance measure. The MSE curves shown in Figure 9 are for the case of an over-determined case where 8 sensors receive signals from 5 sources. Similarly, MSE curves shown in Figure 10 are for the case of an under-determined case where 4 sensors receive signals from 5 sources. As expected, both plots indicate that the proposed method has achieved the best performance for all SNR levels.

Let us now repeat the above experiment sources emitting both amplitude-modulated and frequency-modulated signals.
$$\begin{array}{cc} s_{1}\left(t\right) & =e^{\frac{-|t-64|}{64}}e^{j0.1\pi t+at^{3}}\\ s_{2}\left(t\right) & =e^{\frac{-|t-64|}{64}}e^{j0.8\pi t}\\ s_{3}\left(t\right) & =e^{\frac{-|t-64|}{64}}e^{j0.1\pi t}\\ s_{4}\left(t\right) & =e^{\frac{-|t-64|}{64}}e^{j0.1\pi t-at^{3}}\\ s_{5}\left(t\right) & =e^{\frac{-|t-64|}{64}}e^{j0.5\pi t} \end{array}$$
where $a=4.0690\times 10^{-6}$. The signal is sampled at 1 Hz. We assume that signals are received by 4 sensors and sources are placed at angles 0°, 10°, 20°, 30° and 40°. The TF representations of the signals are given in Figure 11.

The MSE curves are plotted in Figure 12. Simulation results indicate that the proposed method is effective for signals with both frequency modulation and amplitude modulation.

To reproduce the results, code is available from https://github.com/nabeelalikhan1/multi-sensor-IF-estimation, accessed on 21 March 2022.

### 4.2. Interpretation of Obtained Results

The experimental results show that the performance of the proposed method is better than the FAST-IF-based instantaneous frequency estimation method for all SNRs. The proposed method exploits the direction-of-arrival information in addition to chirp rates for accurate tracking of IF curves in the region of intersection that results in better performance. The improved accuracy of the proposed method comes at the expense of a slight increase in computational cost when estimating the spatial frequency ω at the strongest TF point. The computational cost of estimating the IF at the other time instants has not increased. The computational cost of the proposed method is $O(2LKN_{\omega }\log\left(N_{\omega }\right)N_{f}\log\left(N_{f}\right)+6K\Delta W_{l}N_{s})$, where $W_{l}$ is the length of the analysis window, *K* is the number of sources, *L* is the number of quantization levels for estimating the order of fractional Fourier window, $N_{f}$ is the number of frequency bins used estimating the strongest frequency point, $N_{\omega }$ is the number of frequency bins to estimate the spatial frequency ω and $N_{s}$ is the number of samples in the signal. The computational cost of the FAST-IF algorithm is $O(2LKMN_{f}\log\left(N_{f}\right)+6K\Delta W_{l}N_{s})$, where *M* is the number of sensors [2,33].

## 5. Conclusions

A computationally efficient and robust multi-sensor instantaneous frequency estimator has been proposed that exploits the direction-of-arrival information in addition to the rotation order of the fractional Fourier windows for accurate tracking of the IF curves. The ability of the algorithm to exploit the additional information provided by the multiple sensors has resulted in an accurate estimation of IF curves for a signal having little variation in the direction of the IF curve near the intersection region as demonstrated by the experimental results, e.g., the proposed method achieves an MSE of −50 dB at the signal-to-noise ratio of 0 dB, whereas the existing method achieves an MSE of −38 dB. Future work will explore the application of the proposed IF estimator in the reconstruction of multi-sensor sparsely sampled signals [7]. 

## Figures and Tables

**Figure 1 entropy-24-00452-f001:**
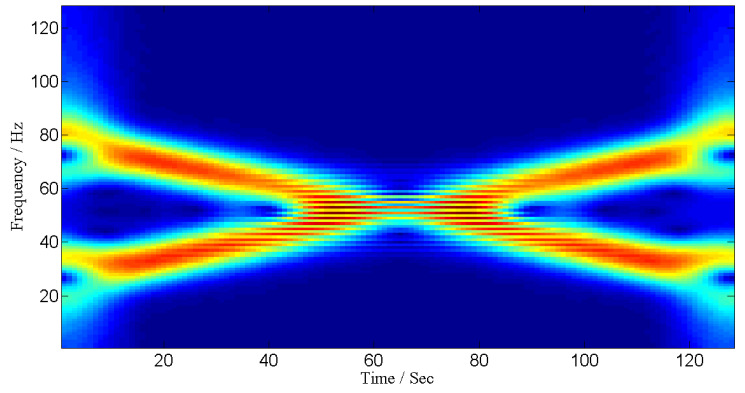
TF representation of a signal in joint TF domain.

**Figure 2 entropy-24-00452-f002:**
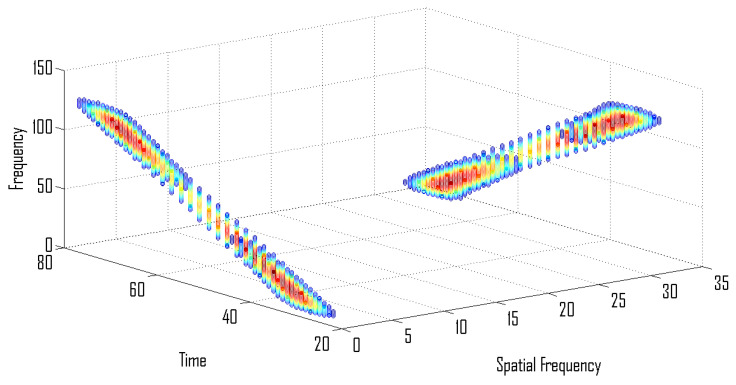
Signal representation in time-frequency–spatial-frequency domain.

**Figure 3 entropy-24-00452-f003:**
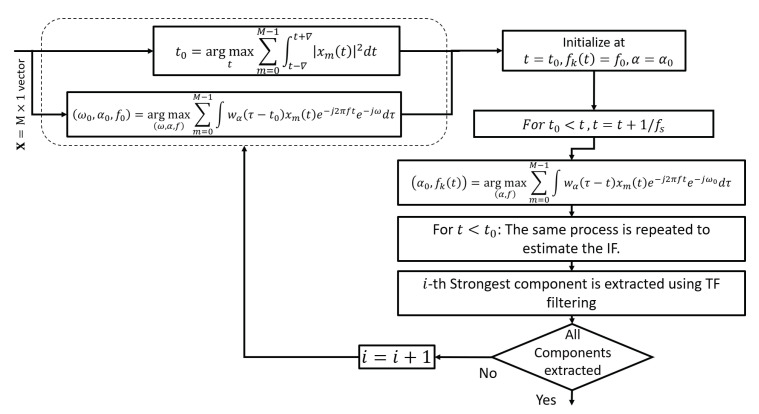
Illustration of the proposed IF estimation scheme.

**Figure 4 entropy-24-00452-f004:**
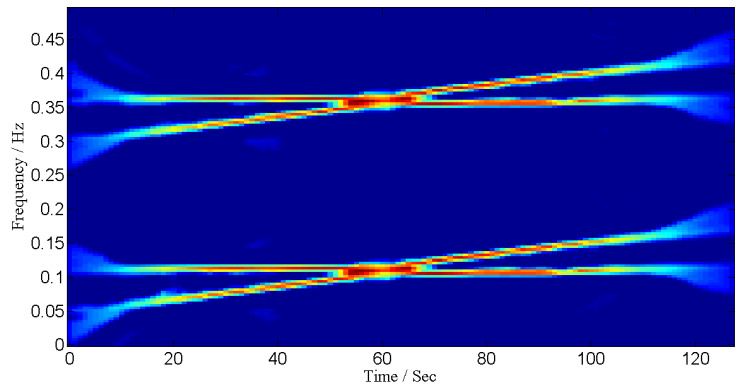
TF representation for linear frequency-modulated signals.

**Figure 5 entropy-24-00452-f005:**
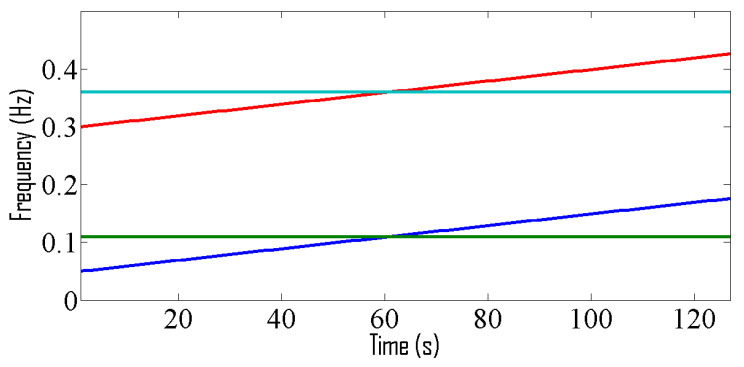
IF curve corresponding to 4 sources emitting linear frequency-modulated signals.

**Figure 6 entropy-24-00452-f006:**
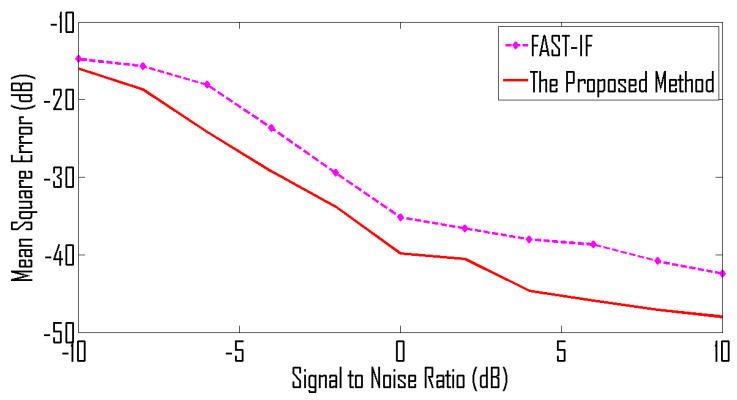
MSE curve for linear frequency-modulated signals received by 8 sensors.

**Figure 7 entropy-24-00452-f007:**
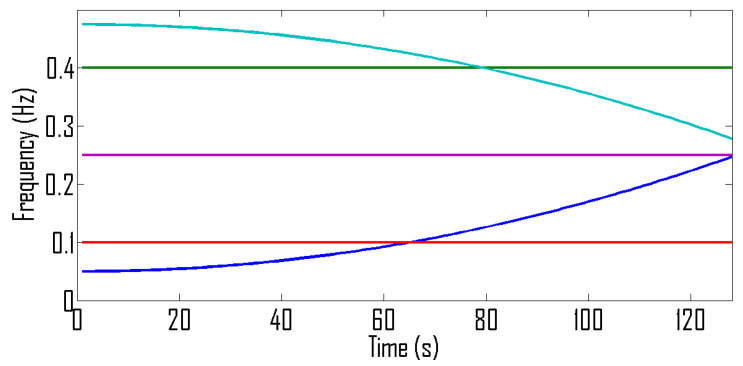
IF curve corresponding to 5 sources emitting non-linear frequency-modulated chirps.

**Figure 8 entropy-24-00452-f008:**
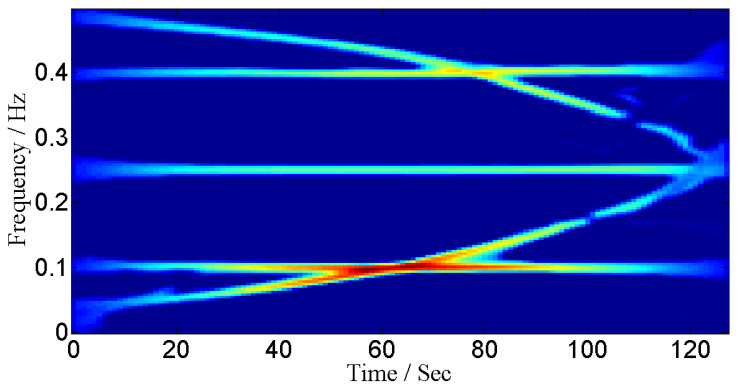
TF representation for non-linear frequency-modulated signals.

**Figure 9 entropy-24-00452-f009:**
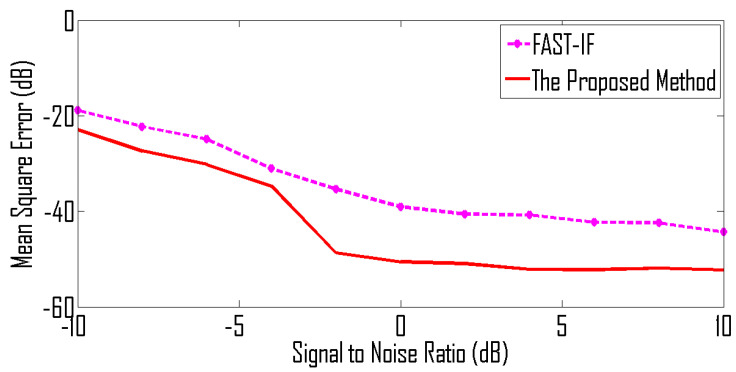
MSE curve for non-linear frequency-modulated signals received by 8 sensors.

**Figure 10 entropy-24-00452-f010:**
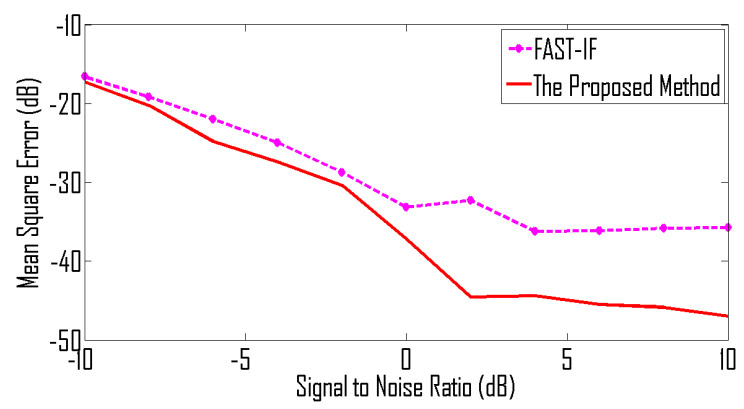
MSE curve for non-linear frequency-modulated signals received by 4 sensors.

**Figure 11 entropy-24-00452-f011:**
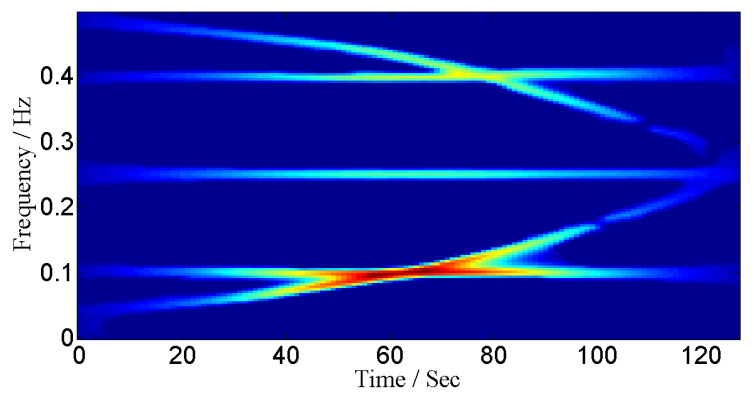
TF representations for amplitude-modulated and frequency-modulated signals.

**Figure 12 entropy-24-00452-f012:**
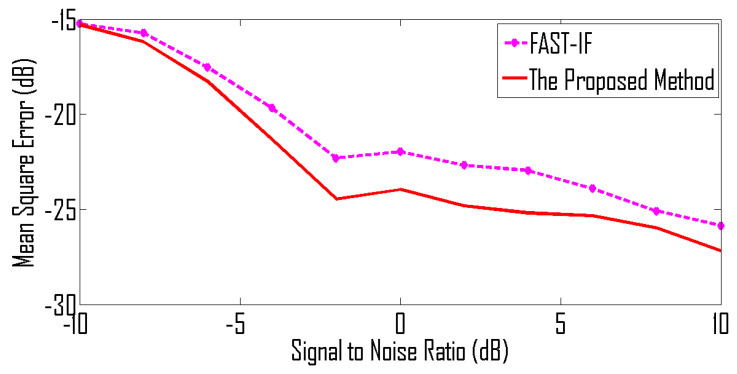
MSE curve for amplitude-modulated and frequency-modulated signals received by 4 sensors.

**Table 1 entropy-24-00452-t001:** Comparison with the state of art.

S.No.	Method	Computational Cost	Limitation
1.	Signal decomposition approaches based on Eigen Vectors [31,32]	High	Applicable only in over-determined scenarios
2.	Spatial TFD methods [3,17]	High	Very high computational cost
3.	FAST-IF [2,33]	Low	Does not take into account spatial frequency
4.	Empirical decomposition method [30]	Moderate	Only applicable to the signals with non-overlapping methods
5.	The Proposed Method	Low	Computational cost is slightly higher than FAST-IF but the method is more robust as it takes into account the spatial frequency

## Data Availability

The relevant code to reproduce results can be downloaded from https://github.com/nabeelalikhan1/multi-sensor-IF-estimation, accessed on 21 March 2022.

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
