# Peer review of "An Efficient and Accurate Multi-Sensor IF Estimator Based on DOA Information and Order of Fractional Fourier Transform"

_entropy, 2022, doi:10.3390/e24040452_

Round 1
Reviewer 1 Report
This paper proposes a method for IF estimation of crossing modes, exploiting both the direction of the IF curve and the angle of arrival, in the multi-sensors framework.
Although the addressed topic seems very interesting to me, in my opinion the article should be improved to be ready for publication. My comments can be found below.
IF estimators or enhancers based on reallocation techniques are not mentioned in the introduction. The same comment applies to hough/radon-transform based approaches and to decomposition schemes for multi-channels multicomponent signals.
Which are the advantages of the proposed method as compared to the methods listed above?
Time-frequency representations of the test signals should be included in the experimental results' section.
Is the proposed method suitable for amplitude-modulated signals?
In addition, is it robust to destructive interference (i.e. significant loss of energy in the TF interference region)?
Is there any relation between the number of sensors and the accuracy in IF estimation?
Author Response
Authors’ Response to Reviewer’ Comments
Manuscript ID: entropy-1617557
Type of manuscript: Article
Title: An Efficient and Accurate Multi-sensor IF Estimator
Authors: Nabeel Khan, Sadiq Ali, Kwonhue Choi *
NOTE: The reviewer issues have been answered in the same order as they were formulated. The number of pages, references, equations and figures are always referred to the original draft unless otherwise specified.
We would like to thank the reviewer and editor for the effort and time devoted in the reviewing process. We sincerely appreciate the comments, the remarks on the original manuscript and the evaluation of the work. They have been very useful to further improve the clarity, completeness and argumentation of the paper. Below, we provide detailed answers to the required changes/modifications and explain how they have been taken into consideration in the revised manuscript.
Reviewer #1 Comments
Comment#1: This paper proposes a method for IF estimation of crossing modes, exploiting both the direction of the IF curve and the angle of arrival, in the multi-sensors framework.Although the addressed topic seems very interesting to me, in my opinion the article should be improved to be ready for publication.
Ans. Thanks for your appreciation. We have addressed all of your observations in the revised manuscript. The changes are highlighted.
Comment#2: IF estimators or enhancers based on reallocation techniques are not mentioned in the introduction. The same comment applies to hough/radon-transform based approaches and to decomposition schemes for multi-channels multicomponent signals.
Ans. In the revised manuscript, we have discussed reassignment methods, Hough-transform-based methods, and multi-channel decomposition methods in the introduction.
Comment#3: Which are the advantages of the proposed method as compared to the methods listed above?
Ans. Time-frequency reassignment methods are post-processing methods that shift the energy of signal energy along the instantaneous frequency curves thus enhancing the readability and resolution of the resultant representation. These methods have been extended for IF estimation of multi-sensor recordings in [1]. The IF in the case of multi-sensor recordings can also be obtained by using multi-variate empirical decomposition methods. However, these methods require signals to have non-overlapping time-frequency signatures [2].
IF estimation problem of signals with overlapping time-frequency signatures can be solved using parametric methods like Hough transform or Ransac based methods. However, these methods approaches are restricted to signals that follow predetermined mathematical models, e.g., LFM signals.
Multi-channel separation methods based on Eigen decomposition are another alternative approach for signal separation. However, this approach requires the number of sources to be more than or equal to the number of sensors.
In summary, the limitations of the aforementioned non-parametric methods that can be employed for signals with overlapping time-frequency signatures are a) these methods are computationally expensive and b) they require a number of signals to be more than a number of sources.
The advantages of the proposed methods as compared to the aforementioned methods are as follows:
- The method is non-parametric, i.e., it does not assume that the IF of the signal follows any mathematical expression.
- The proposed method is computationally efficient as compared to TF-based methods as the proposed method does not require the computation of TFDs of the multi-sensor recordings.
- The proposed method can work in under-determined scenarios.
[1] Synchrosqueezing-based time-frequency analysis of multivariate data
[2] “Empirical Mode Decomposition-Based Time-Frequency Analysis of Multivariate Signals: The Power of Adaptive Data Analysis,
Comment 4: Time-frequency representations of the test signals should be included in the experimental results section.
Ans. Done as suggested.
Comment 5: Is the proposed method suitable for amplitude-modulated signals?
Ans. The proposed method is suitable for signals with amplitude modulation. We have added an additional experiment to illustrate the effectiveness of this approach. Please see Figure 12.
Comment 6: In addition, is it robust to destructive interference (i.e. significant loss of energy in the TF interference region)?
Ans. Yes, the proposed method adopts the order of the fractional Fourier window so that the chirp rate of the analysis window matches with the chirp rate of the component that is being tracked. This avoids destructive interference. This is now discussed in the methodology section.
Comment 7: Is there any relation between the number of sensors and the accuracy in IF estimation?
Ans. Increasing the number of sensors improves the accuracy of the IF estimate, e.g., MSE curves shown in Figure 9 (8-sensors) are lower than MSE curves shown in Figure 10 (4-sensors).
Reviewer 2 Report
Authors presented a novel work.However certain queries/modifications will be required which are as follows :
- Authors are advised to strengthen the abstract and conclusion section. Relevant findings with supporting data are missing.
- Title can be change. Certain keywords related to findings should be included in title.
- Authors can prepare a comparison table with already published literature belonging to same area/methodology to showcase the novelty/utility of methodology proposed.
- Since the findings in submitted manuscript are based on numerical computation and simulation results. Can the proposed methodology be applied to the actual experimental results.
- When multiple sensors mount on the machinery during experimentation,it captures noise also. How robust is proposed algorithm dealing with noise content in signal.
- In page 7,line 104 authors mention about "obtained experimental results" .Kindly correct this sentence.
- Authors are suggested to do proper formatting as some minor errors in formatting are observed.
Author Response
Authors’ Response to Reviewer’ Comments
Manuscript ID: entropy-1617557
Type of manuscript: Article
Title: An Efficient and Accurate Multi-sensor IF Estimator
Authors: Nabeel Khan, Sadiq Ali, Kwonhue Choi *
NOTE: The reviewer issues have been answered in the same order as they were formulated. The number of pages, references, equations and figures are always referred to the original draft unless otherwise specified.
We would like to thank the reviewer and editor for the effort and time devoted in the reviewing process. We sincerely appreciate the comments, the remarks on the original manuscript, and the evaluation of the work. They have been very useful to further improve the clarity, completeness, and argumentation of the paper. Below, we provide detailed answers to the required changes/modifications and explain how they have been taken into consideration in the revised manuscript.
Reviewer 2 Comments
1. Authors are advised to strengthen the abstract and conclusion section. Relevant findings with supporting data are missing.
Ans. We have added quantified results in the Abstract and conclusion sections as suggested.
2. Title can be change. Certain keywords related to findings should be included in title.
Ans. The title has been revised as: “An efficient IF estimator for multi-sensor recordings based on DOA information and order of fractional Fourier transform”.
3. Authors can prepare a comparison table with already published literature belonging to the same area/methodology to showcase the novelty/utility of the methodology proposed.
Ans. Done as suggested. Please see Table 1.
4. Since the findings in submitted manuscript are based on numerical computation and simulation results. Can the proposed methodology be applied to the actual experimental results.
Ans. We have considered realistic signal models of frequency-modulated signals. Such signals are frequently employed in radars, sonars, and communication signals. So, the proposed method remains applicable to real-life signals as well.
5. When multiple sensors mount on the machinery during experimentation, it captures noise also. How robust is proposed algorithm dealing with noise content in signal.
Ans. We have tested the performance of the algorithm for the noisy signal as illustrated by the plots of MSE curves vs. noise in Figures 9, 10 and 12.
6. In page 7,line 104 authors mention about "obtained experimental results" .Kindly correct this sentence.
Ans. Done as suggested.
7. Authors are suggested to do proper formatting as some minor errors in formatting are observed.
Ans. Done as suggested
Reviewer 3 Report
This letter considers the problem of instantaneous frequency estimation. The DOA knowledge provided by additional sensors is exploited to achieve an improved estimation performance in certain settings. I think that the presented idea is interesting and the manuscript is well written. The authors have also posted their code at GitHub to allow the readers reproduce the results. I recommend acceptance of this letter pending some necessary clarifications/revisions.
- I think that the numerical simulations are insufficient. Only one existing method has been compared, which is recently proposed by the authors themselves. The authors are suggested to include more reference methods in the comparison, such as segmented DPT and SDFrFT for multi-component signal analysis, structure-aware Bayesian compressive sensing, optimized sparse fractional Fourier transform, etc.
- Additionally, the simulation settings are too simple to demonstrate the superiority of the proposed method, since the signals to be detected are consist of a few perfect LFM components. If the algorithm is validated using real dataset, the results will be more convincing.
- The authors are recommended to include more theoretical analysis, such as analyzing the theoretical mean-square-error of their method.
- I think that some related works should be mentioned, such as [Ref1]
[Ref1] S. Liu, Y. Ma, and T. Shan, “Segmented discrete polynomial-phase transform with coprime sampling”, The Journal of Engineering, 2019 (19), 5619–5621.
Author Response
Authors’ Response to Reviewer’ Comments
Manuscript ID: entropy-1617557
Type of manuscript: Article
Title: An Efficient and Accurate Multi-sensor IF Estimator
Authors: Nabeel Khan, Sadiq Ali, Kwonhue Choi *
NOTE: The reviewer issues have been answered in the same order as they were formulated. The number of pages, references, equations and figures are always referred to the original draft unless otherwise specified.
We would like to thank the reviewer and editor for the effort and time devoted in the reviewing process. We sincerely appreciate the comments, the remarks on the original manuscript, and the evaluation of the work. They have been very useful to further improve the clarity, completeness, and argumentation of the paper. Below, we provide detailed answers to the required changes/modifications and explain how they have been taken into consideration in the revised manuscript.
Reviewer 3 Comments
Comment#1: This letter considers the problem of instantaneous frequency estimation. The DOA knowledge provided by additional sensors is exploited to achieve an improved estimation performance in certain settings. I think that the presented idea is interesting and the manuscript is well written. The authors have also posted their code at GitHub to allow the readers reproduce the results. I recommend acceptance of this letter pending some necessary clarifications/revisions.
Ans. Thanks for appreciation. We have revised the paper as per reviewer comments.
- I think that the numerical simulations are insufficient. Only one existing method has been compared, which is recently proposed by the authors themselves. The authors are suggested to include more reference methods in the comparison, such as segmented DPT and SDFrFT for multi-component signal analysis, structure-aware Bayesian compressive sensing, optimized sparse fractional Fourier transform, etc.
Ans. We have made the comparison with the method of a similar class, i.e., the one with similar computational complexity and with the ability to estimate the IFs of both linear and non-linear frequency modulated signals. For completeness, we have added the following discussion along with supported references in the introduction to highlight the advantages of our approach with respect to the state of art.
“Discrete polynomial transform and fractional Fourier transform are alternative approaches to estimate the parameters of frequency-modulated signals as discussed in [26-28]. However, these methods are applicable to linear frequency modulated signals only. The method proposed in this study can be applied to a large class of signals as any particular mathematical form is not assumed.”
- Additionally, the simulation settings are too simple to demonstrate the superiority of the proposed method, since the signals to be detected are consist of a few perfect LFM components. If the algorithm is validated using real dataset, the results will be more convincing.
Ans. In section 4.1.2, we have considered examples of signals that are composed of very close and overlapping linear frequency modulated components and non-linear frequency modulated components. In the revised version of the paper, we have considered an additional examples of amplitude modulated and frequency modulated signals (please see figures 11, 12). The signal models used in this study are realistic for a large class of signals.
- The authors are recommended to include more theoretical analysis, such as analyzing the theoretical mean-square-error of their method.
Ans. We have developed an algorithm for a very complicated example of multi-component and multi-channel signals. We assume no particular mathematical form of the given signals. The performance of the algorithm depends on a large number of factors like the number of components, the closeness of signal components, and the number of sensors. So, it is not possible to derive a mathematical expression for this class of signals.
- I think that some related works should be mentioned, such as [Ref1]
[Ref1] S. Liu, Y. Ma, and T. Shan, “Segmented discrete polynomial-phase transform with coprime sampling”, The Journal of Engineering, 2019 (19), 5619–5621.
Ans. Done as suggested.
Round 2
Reviewer 1 Report
The paper has been revised according to my comments and the clarity of presentation has been improved. I only have two minor comments that need very little time to be addressed:
- Caption of figure 4: "non-linear" should be "linear"
- Page 2 lines 24-25: authors stated that [12] assumed linear FM modes. It seems to me that this paper proposed a non-parametric method (but limited to constant-amplitude signals), and both polynomial and hyperbolic FM components are considered in the experimental results. Please, check.
Author Response
Authors’ Response to Reviewer’ Comments
Manuscript ID: entropy-1617557
Type of manuscript: Article
Title: An Efficient and Accurate Multi-sensor IF Estimator
Authors: Nabeel Khan, Sadiq Ali, Kwonhue Choi *
NOTE: The reviewer issues have been answered in the same order as they were formulated. The number of pages, references, equations and figures are always referred to the original draft unless otherwise specified.
Reviewer 1 Comments
- Caption of figure 4: "non-linear" should be "linear"
Ans. Done as suggested
- Page 2 lines 24-25: authors stated that [12] assumed linear FM modes. It seems to me that this paper proposed a non-parametric method (but limited to constant-amplitude signals), and both polynomial and hyperbolic FM components are considered in the experimental results. Please, check.
Ans. Thanks for pointing out the lack of precision in the discussion. We have now rephrased that paragraph for better clarity.
-
Reviewer 3 Report
The authors have adequately addressed all my concerns. I think that the manuscript in its present form is ready for publication.
Author Response
Authors’ Response to Reviewer’ Comments
Manuscript ID: entropy-1617557
Type of manuscript: Article
Title: An Efficient and Accurate Multi-sensor IF Estimator
Authors: Nabeel Khan, Sadiq Ali, Kwonhue Choi *
NOTE: The reviewer issues have been answered in the same order as they were formulated. The number of pages, references, equations and figures are always referred to the original draft unless otherwise specified.
Reviewer 3 Comments
The authors have adequately addressed all my concerns. I think that the manuscript in its present form is ready for publication.
Ans: We would like to thank the reviewer for the effort and time devoted to the reviewing process. We sincerely appreciate the comments in the previous iteration, the remarks on the original manuscript, and the evaluation of the work.